# AlgoLabel: A Large Dataset for Multi-Label Classification of Algorithmic Challenges

**Radu Cristian Alexandru Iacob \*, Vlad Cristian Monea, Dan Rădulescu, Andrei-Florin Ceapă, Traian Rebedea and Ștefan Trăușan-Matu**

Department of Computer Science and Engineering, Faculty for Automatic Control and Computers, University Politehnica of Bucharest Splaiul Independentei 313, Sector 6, 060042 Bucharest, Romania; vlad_cristian.monea@stud.acs.upb.ro (V.C.M.); danradulescu196@gmail.com (D.R.); andrei781995@gmail.com (A.-F.C.); traian.rebedea@upb.ro (T.R.); stefan.trausan@upb.ro (Ș.T.-M.)

\* Correspondence: radu.iacob@upb.ro

**Abstract:** While semantic parsing has been an important problem in natural language processing for decades, recent years have seen a wide interest in automatic generation of code from text. We propose an alternative problem to code generation: labelling the algorithmic solution for programming challenges. While this may seem an easier task, we highlight that current deep learning techniques are still far from offering a reliable solution. The contributions of the paper are twofold. First, we propose a large multi-modal dataset of text and code pairs consisting of algorithmic challenges and their solutions, called AlgoLabel. Second, we show that vanilla deep learning solutions need to be greatly improved to solve this task and we propose a dual text-code neural model for detecting the algorithmic solution type for a programming challenge. While the proposed text-code model increases the performance of using the text or code alone, the improvement is rather small highlighting that we require better methods to combine text and code features.

**Keywords:** text classification; code labeling; multi-modal dataset; multi-label classification; deep learning

---

## 1. Introduction

Recent years have seen an increased interest in semantic parsing, especially due to the advances of data-driven methods using large corpora and deep learning architectures [1,2]. However, in addition to semantic parsing which has been an important Natural Language Processing (NLP) task for decades, several new studies aim to generate complex snippets of code, such as Python or C++, directly from natural language [3,4]. While semantic parsing and code generation are similar, we consider that there are several important differences mainly related to the complexity of the artificial language that needs to be generated. Semantic parsing is aimed to generate queries or logical forms that have a simpler representation or artificial language. At the same time, code generation requires a more complex representation using a programming language that has not only a more complex syntax, but also a larger number of tokens and very difficult semantics and high level programming constructs.

We consider that in order to be able to efficiently generate code from natural language, it is first important to solve some intermediate tasks related to high level programming constructs, such as algorithmic thinking, data structures, and algorithm design techniques. To this extent, a first step is to be able to understand the algorithmic solution required to solve a programming challenge. Thus, we define a multi-label classification task using a large set of challenges gathered from several relevant online resources

for competitive programming. We introduce AlgoLabel, a multi-modal text-code dataset that contains both problem statements and C++ code snippets with solutions for the problems. This dataset can be employed for tagging programming statements with the correct algorithmic solution using the text and code, but also for more complex semantic parsing using real-world problem statements and code snippets.

Our main contributions are twofold. First, AlgoLabel is larger than existing datasets for this task [5] and it has been carefully curated to a small number of classes that are balanced and with data splits employing iterative stratification [6]. Second, it is a multi-modal text-code dataset and we show that a dual text-code classifier achieves better results than text or code alone. We hope that the AlgoLabel dataset and the proposed multi-modal solution will open up new research directions in multi-modal text-code research.

The paper continues as follows. Section 2 contains an overview of the most promising directions in code generation from text and other related tasks using text-code datasets. In Section 3 we define the proposed multi-label classification task using both problem statements and code snippets, continuing with a detailed description of the AlgoLabel dataset in Section 4. The proposed text-code multi-modal architecture, called AlgoLabelNet, for predicting the solution of an algorithmic challenge is presented in Section 5. Section 6 presents the performance of AlgoLabelNet compared with several other strong baselines, while Section 7 provides a discussion on the current limitations and possible improvements.

## 2. Related Work

### Code Generation

The domain of code generation refers to converting natural language descriptions to executable logical forms. We may differentiate between existing challenges in the field based on the complexity and generality of the logical forms and the level of abstraction reached in the natural language descriptions.

We would like to identify two datasets that are directly related to our task: AlgoLISP [2] and NAPS [7]. AlgoLISP leverages algorithmic challenges, automatically synthesized from a small set of computer science homework assignments. The aim of AlgoLISP is to test the ability of learning to compose basic programming routines from simple instructions. Since they are automatically generated, the descriptions exhibit a limited vocabulary and variability. NAPS features problem solutions from programming competitions. The associated natural language statements, obtained via crowd-sourcing, directly specify the precise order in which code instructions have to be called.

Other manually annotated datasets, such as Django [4] and CoNaLa [3], tackle more general programming tasks (e.g., I/O operations, graph plotting, interactions with the OS). Notably, the annotations follow a similar imperative structure, describing the methods that need to be called and their associated arguments. Alternatively, large collections of code-descriptions pairs can be obtained automatically, by scraping open source code repositories [8,9]. However, while the code snippets obtained with this approach can be arbitrarily complex, the descriptions tend to be vague or incomplete.

On the other hand, statements from programming competitions, focus on comprehensively presenting the tasks themselves instead of the steps needed to solve them. Consequently, while this makes them significantly more challenging to understand, even for experienced human programmers, it also provides a more realistic reflection of real-world use cases.

### Semantic Code Representations

Several methods have been published for computing *code embeddings*—continuous vectors which encapsulate the semantics of a code snippet. Research in this area has been driven by the possibility of improving downstream tasks such as automatic code review, API discovery [10] or detecting encryption

functions in malware [11]. Code2Vec [10] extracts paths from the abstract syntax tree (AST) of the source code. These paths are then merged into a single distributed representation by using an attention mechanism. SAFE [11] learns Word2Vec [12] representations for assembly (byte-code) instructions. The instructions corresponding to a function are then reduced to a vector representation via a self-attentive network [13]. Recently, methods which leverage transformers for obtaining contextualized embeddings for code have also been explored [14,15]. Remarkably, transformer-based architectures that have shown very good performance on translation tasks between natural languages [16] can substantially outperform rule-based systems when trained to convert a code snippet from one programming language to another [17].

Multi-modal approaches have also been explored, but to a lesser extent. They have an important advantage as they can provide a method to encode both code snippets and text descriptions to enable applications such as source code retrieval and source code captioning [18,19]. These multi-modal models may be jointly trained to generate natural language summaries of code and code snippets from natural language queries, improving performance on both tasks tackled separately [20].

Algorithm Label Prediction

Investigations into the landscape of competitive programming have revealed that the difficulty of the proposed algorithmic challenges has been consistently increasing [21,22]. The challenge of predicting algorithm labels from natural language descriptions has been introduced recently [5,23]. The task is treated as a multi-label classification problem because algorithmic challenges may have multiple relevant labels as detailed in Section 3. AlgoLabel improves on prior work by also tackling two related tasks: providing algorithmic labels for solution implementations and pairs comprising both the problem statement and corresponding solutions.

The problem of classifying text with multiple labels has received significant attention from researchers [24]. Modern solutions tackle this problem by building complex, deep neural networks [25–27]. One of the latest developments in the NLP community has been to leverage pre-trained Bidirectional Encoder Representations from Transformers (BERT) [28] for downstream tasks, including multi-label document classification [29].

## 3. Task Definition

We ground our exploration with three classification experiments. The first experiment is to assign one or more labels to a programming word problem, given the standard elements received in most competitive programming competitions: natural language description of the statement, the description of the input and output formats and of the time and memory constraints [5]. The second experiment is to classify in the same manner the source code corresponding to a problem solution given as a code snippet. In this formulation we use as inputs data from three different life stages of a solution: the tokens from the original source code, the abstract syntax tree (AST), and its byte-code representation. Finally, we aim to merge the two research threads in a multi-modal setting, by providing annotations for dual pairs consisting of problem statements and their associated code solutions. An example is depicted in Figure 1.

For all experiments, we chose four representative target labels: *math*, *graphs*, *implementation*, and *dynamic programming (dp) & greedy*. The first two tags pertain to the general knowledge required to solve the problem. In particular, we have grouped under the *math* label problems which require specific mathematical insight, such as those already annotated as requiring knowledge of *probabilities*, *number theory*, *game theory*, *geometry*, etc. The *graphs* label describes problems which require modeling the data using a graph structure. Note that the graph is generally not referenced explicitly within the text description. The following tag, *implementation* refers to problems for which the main difficulty relies in converting the abstract solution to actual code. This is a subjective tag which covers a broad selection of problem types:

from simple problems which require the solver to follow a set of instructions (e.g., "simulation"-type problems) to problems covering well-known topics that nevertheless require correct implementation of complex data structures or algorithms. Finally, *dp & greedy* depicts tasks which can be tackled using either dynamic programming or a greedy choice.

The task is inherently a multi-label classification problem as problem statements might have a tag related to the required general knowledge to solve it (e.g., *math*, *graphs*), but also to the method necessary to devise an efficient algorithmic solution (e.g., *implementation*, *dp & greedy*). Nevertheless, there might be statements with only one tag and others with more than two depending on the nature of the problem and on the quality of the tagging process.

**Statement**
You are given a sequence S[] = ($s_1$, $s_2$, .., $s_N$) of size N (1 <= N <= 100000). Compute the maximum sum for a contiguous subsequence in S.

**Input**
On the first line you are given a natural number N. The second line contains N integer numbers, separated by a whitespace, representing the numbers in S.

**Output**
Print a single number representing the sum of the contiguous array within S that has the largest sum.

**Time limit:** 1s
**Memory limit:** 64kb

**Solution**
```
int solve(int[] S, int N) {
    int best = INT_MIN, sum = 0;
    for (int i = 0; i < N; ++i) {
        sum = sum < 0 ? S[i] : sum + S[i];
        best = max(sum, best);
    }
    return best;
}
```

**Labels**: Dynamic Programming (DP)

**Figure 1.** An example of a classic algorithmic challenge. The large upper bound value for *N*, which represents the maximum input size, suggests that an efficient solution needs to employ a dynamic programming technique.

## 4. Dataset

### 4.1. Data Collection

The main difficulty we encountered when building the AlgoLabel corpus was finding open resources which provide a wide range of problem statements from programming competitions, paired with high quality labels and correct solution implementations. One platform which meets all these criteria is Codeforces (https://www.codeforces.com (accessed on 24 June 2020)), a popular online judge which hosts weekly programming contests. These contests range in difficulty, from educational challenges aimed for beginners to particularly difficult tasks designed for skilled coders preparing for competitions.

The data extracted from Codeforces represents the core part of our dataset, with 6374 problems. We filtered problems which required interactions with the online judge and those with an unconventional format (e.g., without a problem statement) or theme (e.g., quantum programming).

Each problem is tagged with labels provided either by the problem writer or by high-rated contestants - all of them can be considered experts or at least highly knowledgeable in algorithms and problem solving. There is however an inherent level of noise in the available labels, which can be explained by the subjective nature of some tags and the possibility of approaching a problem from multiple angles.

We extended the dataset by leveraging Competitive Programming [30], a book which provides a list of high quality label annotations for tasks from various national and international contests. These tasks are hosted on two other popular programming e-judge platforms, Kattis (https://open.kattis.com/ (accessed

on 24 June 2020)) and OnlineJudge (https://onlinejudge.org/ (accessed on 24 June 2020)), and feature a similar format to the ones on Codeforces.

Additionally, for each problem statement from Codeforces we extracted on average 2.48 correct solutions written in C++. Notably, Codeforces competitions test not only the efficiency of the algorithmic approach but also the ability of the contestants to quickly write code. Therefore, submissions often feature macros or other instructions which reduce the size of the code and the time spent on programming. They may also feature unused, pre-written classic algorithms. These approaches tend to affect code readability, adding to the difficulty of applying automated methods to derive meaning from code.

We also collected 22,655 solutions from another online judge, infoarena (https://infoarena.ro (accessed on 24 June 2020)). The majority of submissions extracted from infoarena were not coded within the time constraints of a contest, leading to more readable solutions, with fewer irrelevant snippets. While most problems on infoarena are not labeled, the collected solutions can still be used to improve data driven models on classification tasks through semi-supervised techniques.

Finally, the code corpus also contains 3860 solutions, implemented by university students as programming assignments for the Algorithm Design course at University Politehnica of Bucharest. The students were specifically evaluated on the quality of their coding style, which provided an incentive to write clean, consistent, and well-documented code. We did not include the original problem statements associated to these problems since they were not written in English. All extracted code submissions are written in various dialects of C++.

*4.2. Text Dataset*

In Table 1 we report the number of extracted problems that have associated labels in the AlgoLabel dataset, the average number of labels for each labeled problem, and the number of samples without relevant labels. The latter are provided without a label in order to be leveraged by semi-supervised techniques or to build better language models or representations [12] for this task.

**Table 1.** Statistics for problems from Codeforces (CF), Kattis, and OnlineJudge (OJ).

| Dataset | CF | Kattis | OJ | Total |
|---|---|---|---|---|
| Labeled | 5785 | 771 | 1965 | 8521 |
| Unlabeled | 584 | 1495 | 2935 | 5019 |
| Avg. Num. Labels | 3.19 | 1.72 | 1.78 | |

We compare AlgoLabel with previously published datasets, CFML10 and CFML20 [5] in Table 2. The problems in CFML10 represent a subset of the problems encountered in CFML20, which was extracted from a single source, Codeforces. While in our classification experiments we only leverage 6279 of the total 13,508 samples in AlgoLabel, we believe the remaining examples, both labeled and unlabeled, can drive future exploration in the field.

**Table 2.** Size comparison with previous multi-label datasets for problem tagging prediction.

| Dataset | Size | Num. Labels |
|---|---|---|
| CFML10 [5] | 3737 | 10 |
| CFML20 [5] | 3960 | 20 |
| AlgoLabel | 13,508 | 107 |

Problems from the Codeforces platform have an associated tag which is designed to estimate their difficulty based on the average performance of the participants in contest (For more details, visit codeforces.

com/blog/entry/62865 (accessed on 24 June 2020)). The difficulty tag is a numeric value, ranging from 800 (easiest) to 3500 (hardest). We have considered the problems with a rating less than 1200 to be "easy", between 1200 and 1500 to be "medium", and the remaining problems "hard".

Notably, problems with the *implementation* label appear most frequently among the challenges rated as "easy", while *graphs* are more often encountered among "hard" tasks. We have accounted both for the problem difficulty rating and the overall distribution of labels in the dataset when we have split the data into subsets for training, development, and testing. In order to obtain balanced sets we have applied the iterative stratification technique [6,31]. We included the remaining problems, extracted from other sources than Codeforces, in the training set.

In Table 3, we present general statistics pertaining to the data split for text classification. Each problem sample is separated in three distinct sections: the problem statement and two additional sections depicting in natural language the expected format of the problem input and output. The length and structure of the three sections is consistent across the splits, however the vocabulary encountered during training is remarkably more expansive, accounting for the larger size of the train set.

**Table 3.** Text dataset split statistics.

| Dataset | Train | Dev | Test |
|---|---|---|---|
| dp & greedy | 1866 | 338 | 338 |
| implementation | 1607 | 261 | 261 |
| graphs | 1311 | 208 | 165 |
| math | 1457 | 266 | 266 |
| Unrated Difficulty | 1375 | 0 | 0 |
| Easy Difficulty | 1321 | 243 | 265 |
| Medium Difficulty | 1465 | 358 | 320 |
| Hard Difficulty | 645 | 132 | 155 |
| Size | 4806 | 733 | 740 |
| Avg. Statement Length | 187.3 | 179.2 | 184.2 |
| Avg. Input Length | 75.1 | 63.5 | 66.1 |
| Avg. Output Length | 41.4 | 37.6 | 37.3 |
| Vocabulary Size | 69k | 17k | 17k |

As can be observed from Table 4, *graphs* and *math* problems tend to feature a more specialized vocabulary, with terms that depict relevant concepts appearing more often. Conversely, *implementation* type problems are more general.

**Table 4.** Top 10 words per label, sorted by their TF-IDF value.

| dp & Greedy | Implementation | Graphs | Math |
|---|---|---|---|
| maximum | cards | graph | coordinates |
| a_i | letter | edges | modulo |
| ways | word | roads | point |
| modulo | lowercase | vertex | permutation |
| polycarp | name | edge | answer |
| strings | team | tree | . . . |
| $10^5$ | letters | connected | considered |
| total | characters | u | points |
| array | quotes | road | a_1 |
| choose | table | vertices | f |

*4.3. Code Dataset*

In Table 5 we present general statistics for the extracted code solutions. The code samples from Codeforces feature the most number of solutions, annotated with a difficulty rating and several labels. On Infoarena there are fewer available tags and therefore most submissions, although relevant for this task, are unlabeled. The subset with university assignments has the highest degree of redundancy, with 3860 solutions associated to merely 31 problems.

**Table 5.** Statistics for solutions from Codeforces (CF), Infoarena (IA), and university assignments (Uni).

| Dataset | CF | IA | Uni |
|---|---|---|---|
| Labeled | 15651 | 6912 | 3860 |
| Unlabeled | 171 | 15743 | 0 |
| Avg. Num. Labels | 2.51 | 1.37 | 1.74 |
| Num. Problems | 6374 | 2321 | 31 |
| Avg. Num. Solutions | 2.48 | 9.76 | 124.51 |

When separating the data for the code classification task, in order to allow for a fair comparison between the two research threads, we accounted for the way we performed the split for the natural language classification dataset. Therefore, the subsets used for development and testing contain solutions associated to the same problems from the original text data split. We also added to the training set all the solutions which lacked a difficulty rating.

Although the quality of the coding style varies across the three platforms, the input features we used are designed to mitigate this issue. Thus, we process each source on three different levels of abstraction. The first approach was to extract and anonymize the code tokens. While this is the simplest technique, it is also the most vulnerable to obfuscation and natural variations in implementations, such as the order of instructions. Secondly, we selected paths from the abstract syntax tree, according to the Code2Vec approach for computing semantic representations for code [10]. Finally, we leverage a pre-trained model to derive SAFE embeddings from the source byte-code [11]. With this method we obtain a sequence of distributed representations, one for each compiled function. In Table 6 we capture the average number for these features, as they appear across the data splits.

**Table 6.** Source code dataset split statistics.

| Dataset | Train | Dev | Test |
|---|---|---|---|
| dp & greedy | 7224 | 676 | 654 |
| implementation | 2564 | 528 | 550 |
| graphs | 2873 | 348 | 342 |
| math | 3303 | 576 | 593 |
| Avg. Tokens | 643.5 | 659.4 | 670.8 |
| Avg. Num. AST Paths | 369.4 | 345.5 | 358.0 |
| Avg. AST Path Height | 10.28 | 10.29 | 10.28 |
| Avg. Functions | 3.9 | 3.1 | 3.2 |
| Num. Problems | 3562 | 633 | 633 |
| Size | 14,314 | 1513 | 1513 |

## 5. Methodology

### 5.1. Text Preprocessing

We split each problem statement into sentences and individual words using NLTK [32] and we remove stopwords. Notably, we do not apply lemmatization or stemming as we have observed this hurts the performance of the models.

A particular feature we had to account for was the presence of mathematical formulas used to specify problem input constraints or other relations relevant for the problem statement. First of all, we noticed that mathematical symbols appeared inconsistently (e.g., symbol '$<$' could appear as '\le' or '&lt;'), due to the fact that we have extracted text from different types of sources (html and pdf files). Therefore, we replaced analogous symbols with a unique token.

Numeric constants can provide a useful hint on the expected complexity of the solution and, by extension, the algorithmic technique that needs to be employed. However, there is no meaningful distinction between constants of the same order of magnitude when computing the asymptotic complexity of code. We normalized numeric constants by replacing them with fixed placeholders according to their number of digits.

Moreover, as exemplified in Table 7, we simplify the surface form for three types of formulas that appeared most frequently in the dataset, by grouping together common expressions that shared the same meaning. Thus, we denote the fact that a variable $x$ is defined to have an upper bound limit of $n$ using the expression $range(x, n)$. Likewise, if a sequence $x$ is described textually as having $n$ elements, we replace the snippet with the expression $sequence(x, n)$. We believe that being able to automatically extract this type of information about individual variables can enable more complex reasoning models.

**Table 7.** Examples of formulas with normalized surface form.

| Original Sequence | Normalized Surface Form |
|:---:|:---:|
| $1 \le i, j \le n$ | $range(i, n), range(j, n)$ |
| $0 < n < 200{,}000$ | $range(n, 10^5)$ |
| a_0, a_1, a_2, ..., a_n | $sequence(a, n)$ |
| p_2, \ldots, p_{n} | $sequence(p, n)$ |
| x_1, x_2, ..., x_{n - 1} | $sequence(x, n - 1)$ |
| $(xi, yi)$ | $pair(x_i, y_i)$ |

### 5.2. Code Preprocessing

We used clang-format [33] to normalize the surface form of each code sample. Then we applied cppcheck [34] to statically determine unused functions, which we remove from the representation. Additionally, we eliminate comments and headers. We split the source code into distinct tokens with an open-source code tokenizer [35]. We apply astminer [36] with default parameters to extract AST paths in a format compatible with Code2Vec [10]. We extract at most 400 AST paths for each solution. Afterwards, we compile the solution to byte-code to derive SAFE function embeddings using a pretrained model [11].

### 5.3. Models

Non-Neural Models

Several classifiers that do not leverage neural networks, but that provide good results on textual classification tasks were used: Random Forest [37], SVM [38], and Xgboost [39]. Their input consisted of TF-IDF features computed using scikit-learn [40] and we employed grid search to tune their

hyperparameters. The Random Forest model comprised 200 decision trees with a maximum depth of 150. while the XGBoost solution consisted of 200 trees with a maximum height of 20.

Neural Models

We first trained Word2Vec embeddings [12] on the AlgoLabel training set and on the remaining samples that are not provided with a label. The parameters were initialized using the Xavier method [41] and we used an Adam optimizer [42]. We trained for 10 epochs, with an early stopping mechanism, using mini-batches of 128 samples and cross-entropy loss. We apply L2 regularization and a dropout mechanism to avoid overfitting.

AlgoLabelNet

For the text classification experiments we truncate the size of the problem statement, input, and output sections to 250 tokens. We encode each section separately, using the same bidirectional LSTM [43]. We have augmented the output representation of the encoder using a soft-attention mechanism over the entire input sequence [44]. Next, we concatenate the three inputs and pass the result successively through two fully connected layers. The last layer has a sigmoid activation function for label prediction.

In order to replicate the model for text classification proposed by Athavale et al. [5] we have applied a convolutional neural network on the concatenated input sections. The output is passed through a ReLU activation function, followed by a max pooling layer.

For the code classification experiments we only adapt the encoder to the type of available input features for code. The code tokens are encoded using the same procedure as we did for text, using a bidirectional LSTM, however we set the maximum size to 745. Likewise, we pass the SAFE function embeddings to a distinct bidirectional LSTM, truncated to size 20. The AST paths are aggregated using an attention mechanism, following the methodology proposed in Code2Vec [10].

For all biLSTM models, including AlgoLabelNet, we fix the size of all embeddings, hidden, and fully-connected layers to 100. These hyperparameters were chosen by using the validation set. As an additional neural baseline, we used the BERT base [28] uncased implementation available in the Huggingface library [45].

*5.4. Metrics*

For evaluation, we apply two metrics that are standard for multi-label classification: Hamming loss and the F1 (micro) score. Hamming loss [46] computes the proportion of misclassified examples. Micro-averaged F1 is computed based on the individual true positives, false positives, and false negative values across labels.

## 6. Experimental Results

*6.1. Text Classification*

We capture the performance of the baseline methods in Table 8. Non-neural models leverage bag-of-words features, which cannot fully capture the task complexity. In this experiment, the proposed AlgoLabelNet model achieves the best F1 score of 0.62. However, this still leaves significant room available for improvement. On another hand, the performance of a pre-trained BERT base model fine-tuned on our dataset was poor, with a F1 score of 0.40. We believe this score can be improved with additional parameter tuning on computer science and algorithm books or other similar texts. We also replicated the model proposed by Athavale et al. [5] which uses a CNN encoder and obtained an F1 score of 0.55.

**Table 8.** Performance on the text classification task, measured using Hamming loss (lower is better) and F1 score (higher is better).

| Model | Hamming | F1 |
|---|---|---|
| Random Forest | **0.25** | 0.54 |
| SVM | 0.27 | 0.55 |
| XGBoost | **0.25** | 0.60 |
| BERT | 0.29 | 0.40 |
| CNN [5] | 0.28 | 0.56 |
| AlgoLabelNet (Ours) | 0.27 | **0.62** |
| Ablation study | | |
| - statement | 0.34 | 0.57 |
| - input/output | 0.37 | 0.51 |
| - shared encoder | 0.36 | 0.52 |
| - pretrained embeddings | 0.33 | 0.55 |

Ablation study

We also report the impact of removing either the statement or the input/output sections from the model input. Remarkably, the model performs better with only the input/output format description than when it only receives the actual content of the problem statement. This suggests the model is prone to exploiting language cues (e.g., input size constraints or types of input and output) rather then the text of the problem statement. We explore this issue in more depth in the error analysis section.

In another experiment, we have concatenated the three sections (statement, input, and output) and passed the resulting sequence to a single biLSTM encoder instead of processing them separately. However, this approach proved detrimental to performance for all target labels, yielding an average F1 of 0.52.

Additionally, we measure the impact of improving the word embeddings with data derived from unlabeled problem statements. In this scenario, both metrics degrade significantly. This suggests that learning contextual word embeddings is crucial to getting closer to solving this task.

Label Analysis

In Table 9 we present the precision, recall, and F1 values achieved by XGBoost and AlgoLabelNet for each class. The fact that we obtain better scores for the *graphs* labels is in line with our observation regarding the specialized nature of the vocabulary for these types of problems.

**Table 9.** Text classification performance per class.

| | **XGBoost** | | | **AlgoLabelNet** | | |
|---|---|---|---|---|---|---|
| **Label** | **Precision** | **Recall** | **F1** | **Precision** | **Recall** | **F1** |
| dp & greedy | **0.61** | 0.61 | 0.61 | 0.58 | **0.74** | **0.65** |
| implementation | **0.62** | 0.36 | 0.46 | 0.47 | **0.67** | **0.55** |
| graphs | 0.84 | **0.64** | **0.71** | **0.86** | 0.60 | **0.71** |
| math | **0.72** | 0.53 | **0.61** | 0.59 | **0.62** | **0.61** |

Notably, two models with worse overall F1 scores (Random Forest and XGBoost) achieved a lower Hamming loss than AlgoLabelNet. This can be explained by the fact that the formula for the Hamming loss penalizes equally the situation where a label is wrongly predicted and the case where a correct label is missing from the result. In other words, true positives and true negatives impact the score equally. On the other hand, the F1 metric does not directly account for the number of true negatives encountered.

The samples from the test set feature only 1.39 labels on average from the four chosen target labels. Consequently, a model may trade off recall and F1 score for a better Hamming loss by being biased to assign lower probabilities to all labels. Compared to AlgoLabelNet, XGBoost achieved significantly lower recall for all labels except for the "graphs" category.

### 6.2. Code Classification

For the code classification task we evaluate several baselines and report the results in Table 10. The simplest solution uses a bidirectional LSTM to encode the source code tokens. Despite achieving competitive results in our benchmark, this approach overestimates the importance of variable names. Thus, when changing the original variable names to anonymised placeholders, we notice a significant performance degradation. On our test set, both Code2Vec and SAFE obtain similar scores for the two chosen metrics. Our best neural model, called AlgoCode, merges the two approaches by concatenating the outputs of the two encoders and reaches an F1 score of 0.56. Notably, this score is lower than the one achieved by AlgoLabelNet that uses the textual description of the problems.

Ablation study

We improved the representation of the SAFE embedding encoder by adding an attention mechanism. With this change, the model achieved 0.50 F1 score, from the initial 0.46. We noticed a similar performance gain by varying the maximum number of AST paths for Code2Vec (from 300 to 500, see Table 10).

**Table 10.** Performance on the code classification task, highlighting the different input features: byte-code (BC), AST paths, source code tokens, and anonymised code tokens.

| Model | Input Features | Hamming | F1 |
| --- | --- | --- | --- |
| SAFE | BC | 0.31 | 0.46 |
| SAFE | BC (attention) | 0.35 | 0.50 |
| Code2Vec | AST (300 Paths) | 0.35 | 0.51 |
| Code2Vec | AST (500 Paths) | **0.27** | 0.55 |
| BiLSTM | Code Tok. | 0.33 | 0.55 |
| BiLSTM | Anon. Code Tok. | 0.39 | 0.48 |
| AlgoCode | BC+AST (300)+Tok. | 0.31 | 0.55 |
| AlgoCode | BC+AST (500) | 0.30 | **0.56** |

In Table 11 we present the aggregated code classification performance of the best AlgoCode model for each label. Similar to the natural language challenge, the *graphs* category is the easiest to recognize, while *implementation* lags behind according to our metrics.

**Table 11.** Code classification performance per class for the AlgoCode model.

| Labels | Precision | Recall | F1 |
| --- | --- | --- | --- |
| dp & greedy | 0.52 | 0.61 | 0.57 |
| implementation | 0.53 | 0.53 | 0.53 |
| graphs | 0.68 | 0.60 | 0.64 |
| math | 0.60 | 0.53 | 0.56 |

### 6.3. Dual Text-Code Classification

For this final experiment, we combined our two best models in a single unitary framework. Thus, we have a neural model which takes both the natural language description of a problem and its corresponding coded solution and classifies the problem accordingly into one or more algorithmic labels. As natural

language inputs we use the statement, input and output sections. From the source code, we leverage the SAFE embeddings and the Code2Vec distributed representation. All these inputs are encoded as previously presented, then concatenated and passed to a fully connected feedforward layer.

We restricted the training dataset to only Codeforces problems, since this is the only section of our dataset that features both problems and their solutions. The size of the test and development set remains unchanged, since we keep the same problem distribution from the text classification experiment. For each problem we randomly selected a corresponding solution.

As depicted in Table 12, despite the reduction in the size of the training set, the dual text-code model achieves better aggregated results than the two models evaluated separately. For every label, except for *implementation*, the F1 score improves compared to the results obtained on a single type of input. However, the improvement achieved by the dual model is small compared to AlgoLabelNet, suggesting the code brings little information in addition to the text, at least given the current code features.

**Table 12.** Dual text-code classification performance.

| Model | Input Features | Hamming | F1 |
|---|---|---|---|
| AlgoLabelNet | Text+Code | 0.26 | 0.65 |
| **Labels** | **Precision** | **Recall** | **F1** |
| dp & greedy | 0.55 | 0.81 | 0.65 |
| implementation | 0.58 | 0.56 | 0.57 |
| graphs | 0.72 | 0.75 | 0.74 |
| math | 0.55 | 0.81 | 0.65 |

## 7. Discussion

### 7.1. Text Classification

The thought process behind understanding a problem statement often resembles solving a riddle: in a competition setting, in many cases, the author of the statement tries to hide the problem requirements behind a story. The chosen story may sometimes reference a relevant real-life application.

Thus, the statement often begins with a prologue, which provides the setting of the story. Typically, since it does not reference any constraints, the prologue can be safely ignored, without affecting the ability to understand the problem. This section is followed by sentences that depict the actual requirements. Among these, we may distinguish the ones that surmise the problem objective (e.g., "Find the minimum value for X"). Additionally, the statement may also contain hints regarding the automatic evaluation platform where the problem is hosted on. For example, a problem may include tips related to how to implement a solution in a specific programming language: particular functions that should be called, the fact that the output requires a 64 bit representation, etc. As in the case of the prologue, these sentences provide no insight regarding the solution for the problem to a human reader, however they may confuse a data driven neural model.

Unravelling the relevant facts from the narrative is a non-trivial but essential challenge, particularly for real-world applications. In our experiments, we have observed that the input and output sections, which are typically written in a more formal language, have a larger impact on the classification score than the statement. This finding was also reported in a previous study [5].

Case study: Graph problems

Problems which can be modeled using a graph data structure are typically easier to identify due to the specialized vocabulary used (e.g., explicit references to vertexes or graph edges). However, in situations



where the graph modelling is not obvious from the problem statement, the proposed models struggle to recognize the type of the problem.

In the majority of misclassified graph problems from the test set, the graph is not explicitly provided as input. For example, the statement may describe an initial state (e.g., a string), how to transition from one state to another (e.g., permute the letters in the given string) and enquire about the path that leads to a desired final state. If the problem introduces a novel concept as a state, such as an image, a symbolic expression or a number sequence, the models fail to recognize the abstract nature of the task.

Specialized graph data structures may constitute only an auxiliary component of the solution, used to optimize certain operations. We investigate this scenario in Figure 2, where we encounter the statement of a problem (For more details, visit https://codeforces.com/problemset/problem/566/A (accessed on 24 June 2020)) annotated with the *graph* label by the author. The neural baseline assigns a low probability for the *graph* label (0.19) and a high probability for *dp&greedy* (0.75). The low *graph* probability can be explained by the fact that there is no explicit graph provided as input. Nevertheless, the solution requires the construction of a trie data structure to efficiently store and process the collection of input strings. Instead, guided by the presence of expressions such as "*largest common*" or "*maximum*", the model makes the assumption that this is a combinatorial optimization problem. Although the label is missing from the dataset, the assumption is actually correct in this case: the official solution does indeed apply a greedy algorithm on the trie in order to derive the correct answer.

> Teachers [of] one programming summer school decided [to] make a surprise for [the] students by giving them names in [the] style [of the] " hobbit " movie . each student must get a pseudonym maximally similar his own name . [The] pseudonym must be a name [of] some character [of the] popular [saga and] now teachers are busy matching pseudonyms [to] student names .
>
> There are n students in a summer school . Teachers chose exactly n pseudonyms for them . Each student must get exactly one pseudonym corresponding [to] him . Let us determine [the] relevance [of] a pseudonym b [to] a student [with] name a as length largest common prefix a [and] b . We will represent such value as [lcp(a,b)] . Then we can determine [the] quality matching [of the] pseudonyms students as a sum [of] [relevances] [of] all pseudonyms [to] corresponding students . Find [the] matching between students pseudonyms [with the] maximum quality .

**Figure 2.** Attention scores for the statement of a graph problem that was misclassified as dp & greedy. Stopwords 'the', 'of', 'to' are filtered and every word is converted to lowercase prior to training the model. Remaining words without an associated embedding are marked with brackets.

*7.2. Code Classification*

It may seem easier, given its unambiguous nature, to extract semantic knowledge from code than it is from natural language. However, in practice, approaches that work well for natural languages need to be carefully tailored to account for the specific structure of programming languages. In our code classification experiments, we have encoded the solution as it was represented during three different compilation stages: as a text sequence, as a collection of AST paths, and as byte-code.

As described in Section 5.2, processing the original snippet as a text sequence is laborious, requiring several steps to eliminate uninformative segments from the input. Code styling conventions mandate the need for self-explanatory names for variables, however, in a competition environment, solution authors frequently rely on short, generic names that can be typed faster (e.g., *i, var, res*). The model needs to account for the type as the multiple contexts in which a variable is referenced, in order to understand its role in the solution. Notably, in a programming language that allows access to arbitrary locations in memory, a variable may be referenced indirectly.

The second approach we considered were context paths derived from the abstract syntax tree. These paths are more generic and may be better at capturing the high level structure of the solution. However, a weakness of the current method [10] is that novel AST paths, not available in the training set, can not be represented during test time. Moreover, this approach is sensible to the number of AST paths considered and the maximum height allowed for a path.

Finally, we have encoded byte-code representations of each function using SAFE embeddings. These embeddings were pre-trained on a related but different semantic classification task. We believe we can improve results on our task by pre-training SAFE or a similar model on a specialized labeled dataset comprising algorithmic solutions. At the same time, learning unsupervised contextual code embeddings, as presented in recent literature, by training different transformer models on collections of very large code repositories [14,15,17], may hold the key to improving semantic understanding of code.

An interesting extension to our work would be to encode program execution traces, as suggested by Wang and Su [47]. Exploiting runtime information provides the opportunity to capture more complex program semantics, and thus outperform syntax-based methods. However, this approach requires both a runtime to execute the code as well as access to test input data, which may not always be available and that is not part of the AlgoLabel dataset at this time.

Leveraging the solution, in addition to the problem statement, increases the performance of the classification compared to using the problem statement alone. However, this improvement is small, especially due to the fact that the code classification task has significantly poorer results than the text classification and due to the smaller training dataset for the multi-model text-code dataset than for the unimodal (text or code) datasets.

### 7.3. Computational Efficiency

The memory space used by the neural architectures that we have experimented with is independent of the input size, being determined by a small set of hyper-parameters: the size of hidden state for the LSTM encoder, the number of filters and their size for the CNN module and the size of the fully connected layer used to transform each AST path for the Code2Vec attention method. Regarding the time complexity, we can identify precise asymptotic upper-bounds for training the non-neural baselines. On the other hand, for the solutions employing neural networks it is difficult to predict the number of training steps required to achieve good performance.

From a practical perspective, the main resource consumption difference between the specified encoders is the potential for parallelization: the LSTM module has to scan the input sequentially, while the other encoders apply independent computations that may be performed efficiently in parallel.

### 7.4. Study Limitations

Annotation issues

The labeling of problem statements were provided by their authors and field experts. However, there are several issues that may be identified with the annotation process. First of all, given the number of people involved and the subjective nature of some labels, such as *implementation*, it is possible for labeling inconsistencies to occur in the dataset. Thus, different annotators may select different labels as relevant for a given problem, even while considering the same solution. Moreover, problems may also accept multiple, fundamentally different solutions. Finally, we have also identified few erroneously added labels in our dataset.

Explainability

The proposed neural models are only able to highlight the key words or phrases that have determined a particular prediction. We cannot use them to automatically infer why a particular phrase is relevant. Moreover, solving hard problems requires several reasoning steps. Identifying these steps goes beyond analysing the surface description of the problem statement.

Data variation

Another issue with the present experiment is the small number of labels considered. A more fine-grained study, covering rare labels, could provide valuable insights. Finally, the proposed dataset is still relatively small, which may prevent more data-hungry models, such as BERT, from leveraging it successfully.

## 8. Conclusions

In this paper we have introduced AlgoLabel, a new multi-modal dataset comprising problem statements for algorithmic challenges and their associated code solutions. We believe this new resource will encourage further research in extracting algorithmic knowledge from text, a necessary stepping stone towards general semantic parsing. A tool that can recognize the problem requirements from an informal story, and thus help design a solution, may prove of significant value, even to an experienced programmer. Moreover, we have showed that efforts towards deriving semantic understanding from either text and code can benefit from jointly modeling data from the two domains.

Furthermore, we have investigated several baselines for the task of multi-label classification of problem statements (AlgoLabelNet) and their corresponding source code (AlgoCode). AlgoLabelNet leverages a biLSTM encoder model to separately encode the three sections of a problem statement, with word embeddings pre-trained on a larger collection of unlabeled algorithmic challenges, provided in our dataset. For the code experiments, we have captured the solution representation from three different compilation stages and explored the advantages and disadvantages of each representation. Additionally, we have experimented with a dual text-code neural model, which achieves improved performance over considering the code or the text alone.

**Author Contributions:** Conceptualization: R.C.A.I. and T.R., Methodology: R.C.A.I. and T.R., Software: R.C.A.I., V.C.M., D.R. and A.-F.C., Validation: R.C.A.I., V.C.M., D.R. and A.-F.C., T.R. and Ș.T.-M. Writing: R.C.A.I., T.R. and Ș.T.-M., Supervision: T.R. and Ș.T.-M. All authors have read and agreed to the published version of the manuscript.

**Funding:** This research received no external funding.

**Conflicts of Interest:** The authors declare no conflict of interest.

**Data Availability:** The AlgoLabel dataset and the experiment code are available online at https://github.com/NLCP/AlgoLabel.

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
