# Peer review of "AlgoLabel: A Large Dataset for Multi-Label Classification of Algorithmic Challenges"

_mathematics, doi:10.3390/math8111995_

Round 1
Reviewer 1 Report
The main criteria for review can include the followings:
The main idea of the paper:
Multi modal dataset of problem statements for algorithmic challenges and their associated code solutions.
Provides a stepping stone with neural networks towards general semantic parsing.
1. Originality and significance of the research
Very original in its scope, well defined and significant for eventual automatic algorithmic knowledge extraction.
2. Technical and theoretical correctness
Technically correct, theoretically correct.
3. Readability of the paper
Very readable. Good English.
Is this accessible?
http://codeforces.com/problemset/problem/566/A
Doesn't work on my browser
4. Evaluation result
Semantic classification task.
Result shows significant improvement on F1. Could you elaborate more on the significance of your Hamming result?
Overall good.
5. Scope of the work.
Acceptable within scope.
Author Response
Thank you for your review!
- We are sorry, the “https” protocol was missing from for the URL; we have edited it, it should be accessible now.
- Thank you for your suggestion. In our text classification experiments, we encountered an interesting situation where two models with worse overall F1 scores (Random Forest and XGBoost) achieved a better, lower Hamming loss than AlgoLabelNet.
We have added the following paragraphs in section 6.1 “Text Classification”:
“Notably, two models with worse overall F1 scores (Random Forest and XGBoost) achieved a lower Hamming loss than AlgoLabelNet. This can be explained by the fact that the formula for the Hamming loss penalizes equally the situation where a label is wrongly predicted and the case where a correct label is missing from the result. In other words, true positives and true negatives impact the score equally. On the other hand, the F1 metric does not directly account for the number of true negatives encountered.
The samples from the test set feature only 1.39 labels on average from the four chosen target labels. Consequently, a model may trade off recall and F1 score for a better Hamming loss by being biased to assign lower probabilities to all labels. Compared to AlgoLabelNet, XGBoost achieved significantly lower recall for all labels except for the “graphs” category.”
Reviewer 2 Report
I appreciate the discussed results. The paper contains the comparisons needed to approve the problematic announced by the authors in the introduction section. The work can be published in MDPI Mathematics.
Author Response
Thank you for your review!
Reviewer 3 Report
This is a very thorough research that propose an alternative problem to code generation: labelling the algorithmic solution for programming challenges. I read the manuscript with great interest and believe its topic is important and relevant. The authors performed a careful and thorough review of the literature, as the section was very informative and substantial. Appropriate theoretical framework was applied. I found the methodological part to be well justified and reasonable for this type of analysis. Although the manuscript is overall well-written and structured, it might benefit from additional spell/language checking. However, I have some comments which I would like to be addressed before the acceptance of this paper.
Comments
- What was the key motivation behind proposing the AlgoLabel?
- Why author’s used Multi-label Text Classification?
- What are some key issues that present study has addressed?
- Make comparison of your study findings with the past studies that used same dataset in discussion section. And in this context it is worth mentioning their experimental evaluation protocol for a fair comparison.
- The AlgoLabelNet in Table 8. Showed score of F1 0.62, but would be interesting if the authors report the trade-off compared to other methods especially the computational complexity of the models. Some techniques require more memory space and take longer time, please elaborate on that.
- What are the limitations of your study?
I look forward to read the final version.
Author Response
Thank you for your helpful comments.
- What was the key motivation behind proposing the AlgoLabel?
- Why author’s used Multi-label Text Classification?
- What are some key issues that present study has addressed?
The main motivation behind AlgoLabel is to merge and advance research in two related strands of work: deriving semantic knowledge from algorithmic problems, described using natural language and their associated solutions.
Current state of the art in the field of code generation from text is limited to relatively straightforward tasks, where the target instructions are provided or referenced explicitly. Parsing an algorithmic problem is significantly more challenging, since the solver needs to reason to identify the problem objective and how to tackle it efficiently. Conversely, as our experiments attest, correctly identifying the programming paradigms and techniques from code is a difficult task on its own. Thus, we pursue a multi-label classification task for text and code, as a stepping stone towards the general goal of semantic parsing. To our knowledge, this is the first dual-modal dataset used to pursue this objective.
These issues were addressed and expanded in the Introduction and Related Work section.
- The AlgoLabelNet in Table 8. Showed score of F1 0.62, but would be interesting if the authors report the trade-off compared to other methods especially the computational complexity of the models. Some techniques require more memory space and take longer time, please elaborate on that.
The computational complexity of the models we experimented with is detailed in this answer. In the current version of the paper, we have added a shorter explanation in Section 7.3 as follows below. If the reviewer considers that this clarification is too short, we can add some or the entire discussion presented below.
“The memory space used by the neural architectures that we have experimented with is independent of the input size, being determined by a small set of hyper-parameters: the size of hidden state for the LSTM encoder, the number of filters and their size for the CNN module and the size of the fully connected layer used to transform each AST path for the Code2Vec attention method. Regarding the time complexity, we can identify precise asymptotic upper-bounds for training the non-neural baselines. On the other hand, for the solutions employing neural networks it is difficult to predict the number of training steps required to achieve good performance.
From a practical perspective, the main resource consumption difference between the specified encoders is the potential for parallelization: the LSTM module has to scan the input sequentially, while the other encoders apply independent computations that may be performed efficiently in parallel.“
Non-Neural Models
Model |
Space |
Train |
Prediction |
Random Forest [0] |
O(T * D) |
O(T * K * N * D) |
O(T * D) |
XGBoost [1] |
O(T * D) |
O (T * D * X + X * log(N)) |
O(T * D) |
SVM [2] |
O(N^2) |
O (N^3) |
O(V * F) |
N - number of examples
T - number of trees
D - maximum depth
K - the number of features considered when looking for the best split on each level of a decision tree. The default value for K in the Scikit implementation is √F, where F stands for the total number of features. In our experiments, F is equal to the size of the vocabulary.
X - number of non-missing entries in the training data
V - number of support vectors
[0] Louppe, G. (2014). Understanding random forests. Cornell University Library.
[1] Chen, T., & Guestrin, C. (2016, August). Xgboost: A scalable tree boosting system. In Proceedings of the 22nd acm sigkdd international conference on knowledge discovery and data mining (pp. 785-794).
[2] Abdiansah, A., & Wardoyo, R. (2015). Time complexity analysis of support vector machines (SVM) in LibSVM. International journal computer and application, 128(3), 28-34.
Neural Models
For the neural models we have experimented with, the component that is most computationally intensive and takes up the most resources is the input encoder.
The LSTM based encoder contains four units: the input gate, the forget gate, the cell state and the output gate. Each unit comprises a matrix that transforms the input (the kernel), the hidden state from the previous time step (the recurrent kernel) and a bias component. Therefore, the size of the LSTM encoder is 4 * (HS * ES + HS * HS + HS) , where HS is a hyperparameter which denotes the size of the hidden state and ES is the size of the input embedding. We share the same LSTM weights for processing each input section.
The CNN based encoder processes the input sequence using NF filters. The space used is NF * FS * ES, where FS is a hyperparameter describing the dimensionality of the filter.
The Code2Vec type encoder transforms each AST path independently using a fully connected layer, before applying an attention mechanism to determine the most relevant inputs.
From a practical perspective, the main resource consumption difference between the specified encoders is the potential for parallelization: the LSTM module has to scan the input sequentially, while the other encoders apply independent computations that may be performed efficiently in parallel.
- Make comparison of your study findings with the past studies that used same dataset in discussion section. And in this context it is worth mentioning their experimental evaluation protocol for a fair comparison.
A related, multi-label experiment, only for natural language input, was performed in [0], although on a different dataset (a subset of AlgoLabel) and a different set of target labels. The performance was measured using the same metrics, Hamming loss and F1 score. The proposed neural architecture, which uses a CNN encoder, was included in our own experiment. We have highlighted a common observation with this study in subsection 6.1.
[0] Shrivastava, M. (2019, November). Predicting Algorithm Classes for Programming Word Problems. In Proceedings of the 5th Workshop on Noisy User-generated Text (W-NUT 2019) (pp. 84-93).
- What are the limitations of your study?
Thank you for your suggestion, we have added a new subsection, 7.4, where we enumerate several study limitations.
Round 2
Reviewer 3 Report
All my comments are addressed hence; manuscript is accepted.